# Chemical, Thermal, Time, and Enzymatic Stability of Silk Materials with Silk I Structure

**DOI:** 10.3390/ijms22084136

**Published:** 2021-04-16

**Authors:** Meihui Zhao, Zhenzhen Qi, Xiaosheng Tao, Chad Newkirk, Xiao Hu, Shenzhou Lu

**Affiliations:** 1National Engineering Laboratory for Modern Silk, College of Textile and Clothing Engineering, Soochow University, Suzhou 215123, China; 18340837762@163.com (M.Z.); 20204215016@stu.suda.edu.cn (Z.Q.); taoxiaosheng520@163.com (X.T.); 2Department of Physics and Astronomy, Rowan University, Glassboro, NJ 08028, USA; newkir16@students.rowan.edu; 3Department of Molecular and Cellular Biosciences, Rowan University, Glassboro, NJ 08028, USA

**Keywords:** silk fibroin, porous material, crystalline structure, stability

## Abstract

The crystalline structure of silk fibroin Silk I is generally considered to be a metastable structure; however, there is no definite conclusion under what circumstances this crystalline structure is stable or the crystal form will change. In this study, silk fibroin solution was prepared from B. Mori silkworm cocoons, and a combined method of freeze-crystallization and freeze-drying at different temperatures was used to obtain stable Silk I crystalline material and uncrystallized silk material, respectively. Different concentrations of methanol and ethanol were used to soak the two materials with different time periods to investigate the effect of immersion treatments on the crystalline structure of silk fibroin materials. X-ray diffraction (XRD), Fourier transform infrared spectroscopy (FTIR), Raman scattering spectroscopy (Raman), Scanning electron microscope (SEM), and Thermogravimetric analysis (TGA) were used to characterize the structure of silk fibroin before and after the treatments. The results showed that, after immersion treatments, uncrystallized silk fibroin material with random coil structure was transformed into Silk II crystal structure, while the silk material with dominated Silk I crystal structure showed good long-term stability without obvious transition to Silk II crystal structure. α-chymotrypsin biodegradation study showed that the crystalline structure of silk fibroin Silk I materials is enzymatically degradable with a much lower rate compared to uncrystallized silk materials. The crystalline structure of Silk I materials demonstrate a good long-term stability, endurance to alcohol sterilization without structural changes, and can be applied to many emerging fields, such as biomedical materials, sustainable materials, and biosensors.

## 1. Introduction

Silk fibroin is a natural biopolymer that can be regenerated and processed into various material forms, such as films, sponges, hydrogels, and microspheres, and has broad application prospects in the field of biomaterials [1,2,3]. However, the various aggregated structures of silk fibroin have not been clearly revealed for decades. The main body of the silk fibroin chain segment is formed by alternating blocks of crystalline and amorphous regions [4,5]. Among them, the crystalline region is dominated by the GAGAGS (amino acids Gly-Ala-Gly-Ala-Gly-Ser) sequence with short side chains. The arrangement of amino acid residues in the amorphous region is complex and contains many amino acid residues with long side chains, such as tyrosine, lysine, and arginine. These residues are relatively hydrophilic and hinder the regular aggregation and crystallization of the chain segments, resulting in a random coiled molecular conformation [6,7]. Research on the crystalline form of silk fibroin can be traced back to Shimizu Masanori, who discovered the first two crystalline forms of silk fibroin, named silk fibroin α and β [8]. Later, two types of crystals of silk fibroin, Silk I and Silk II, were discovered [9]. The Silk I crystal structure model is a repeating unit dipeptide, with a molecular chain in the shape of a crankshaft. Silk I is neither an α-helical structure nor a β-sheet structure, and it belongs to the orthorhombic crystal system [10]. The Silk II crystal structure model is a layered structure formed by β-antiparallel folding, belonging to the monoclinic crystal system, with the strength, toughness, and anti-solubility of silk fibroin [11]. Valluzzi et al. [12] discovered a new crystalline form at the interface between silk fibroin solution and air, called Silk III, a structure similar to a 3-fold helix. Silk III is similar to polyglycine II and is a member of the hexagonal crystal system. The performance of the silk fibroin materials is strongly influenced by the individual crystal system.

Studies on the crystalline structure of Silk I is slowly developing. Valluzzi et al. and Lu et al. demonstrated that adding polyol [13] with a specific balance of hydrophilic and lipophilic properties while slow drying [14] under controllable relative humidity and temperature conditions can induce the formation of Silk I. Silk I is a hydrated structure: increasing the osmotic pressure of the solution can agglomerate silk fibroin macromolecules to form a Silk I structure [15]. Compared to materials with Silk II crystalline structure, Silk I crystalline material shows a higher flexibility and faster degradation rate [7,10]. Lu S [16] inserted the prepared porous material Silk I into the subcutaneous layer on the dorsal surface of SD rats, and all the rats could survive and live in good condition. After 6 weeks, the porous material Silk I was degraded and absorbed by the rats at most. Hu X et al. demonstrated a new physical approach to control the structure of fibrous proteins through temperature-controlled water vapor annealing (TCWVA), making it a suitable starting material for any complex format, such as patterned films, foams, nanofibers, micro/nano-particles, or gels [10]. Jin H effectively controlled the β-sheet ratio to obtain a water-stable film containing Silk I crystals [17]. Zhu M et al. used combined Silk I microneedles for drug delivery of insulin delivery [18]. Previous studies showed that Silk I-based materials had a metastable structure with poor stability and short storage time. In order to make Silk I crystal structure materials more widely used, this paper discusses the long-term stability of Silk I-based materials in detail, which includes water resistance stability, monohydric alcohol treatment stability, time stability, high temperature stability, and enzyme degradation stability, to provide relevant data and promote its applications in the biomedical field.

## 2. Results

### 2.1. Water Resistance Stability

With the extension of the immersion time in water, the dissolution rate of the randomly crimped silk fibroin material (Figure 1) gradually increases, and the dissolution rate increases significantly after 2–12 h. After 12 h, the dissolution loss rate of SF-R (Silk Fibroin Random coil porous material) reached 71%. The dissolution loss rate of SF-I (Silk Fibroin Silk I crystalline structure porous material) remains mostly unchanged with time. After soaking in deionized water for 12 h, the dissolution loss rate is less than 0.48%. This result shows that the Silk I crystalline structure material can exist stably in water without dissolution loss, while the random coil structures in the SF-R sample are lost during the dissolution process. In Figure 1, SF-I represents Silk I crystalline structure porous material, and SF-R represents random coil porous material. The icons in the following text all represent this meaning.

### 2.2. Chemical Stability

#### 2.2.1. X-ray Diffraction Analysis

Previous X-ray diffraction studies showed that the diffraction peaks of silk fibroin Silk I crystals mainly appeared at 12.2°(ms), 19.7°(s), 24.7°(m), 28.2°(m), 32.3°(w), 36.8°(mv), and 40.1°(mw); and the diffraction peaks of Silk II crystal mainly appeared at 9.1°(ms), 18.9°(ms), and 20.7°(vs.) [19,20]. X-ray diffraction analysis in Figure 2 shows that the random coiled porous silk material (SF-R) mainly exhibited an amorphous structure without the appearance of crystalline peaks. After 8 h of treatment with different concentrations of methanol and ethanol, crystallization peaks appeared at 9.1°and 20.7° for the SF-R material (Figure 2a–f). Through this process, the random coil structure was transformed into Silk II crystalline structure. Due to the monohydric alcohols’ capability of entering the interior of the silk fibroin material with ease, the silk fibroin molecules can move more freely, thereby changing to a more stable crystalline structure, causing the silk fibroin to change from an amorphous state to a crystal structure [11]. However, the crystal structure of Silk I-based material (SF-I) remained as Silk I crystal, and no significant difference was observed in the X-ray diffraction (XRD) analysis before and after alcohol immersion treatments. This result indicates that the Silk I crystal structure in the silk fibroin material has good chemical stability and will not be transformed into Silk II crystal structure under the influence of methanol or ethanol.

#### 2.2.2. Fourier Transform Infrared Spectroscopy (FTIR) Analysis

Figure 3 shows the characteristic peaks in the FTIR spectrum of SF-R porous materials at 1644 cm^−1^ (amide I), 1531 cm^−1^ (amide II), and 1235 cm^−1^ (amide III), which illustrated that silk fibroin molecules in SF-R samples are dominated by random coil structures. Extension of the methanol and ethanol immersion treatment time shows the random coil structure gradually transforming into β-sheet structure. Silk I material (SF-I) has infrared absorption peaks near 1651 cm^−1^ (amide I), 1531 cm^−1^ (amide II), and 1234 cm^−1^ (amide III). The absorption peaks shown are different from the β-sheets peaks in SF-R samples. The samples processed for 8 h were selected for infrared study, resulting in the absorption peak positions in the amide I and amide II regions to remain mostly unchanged for the Silk I material (SF-I) with the increase of ethanol and methanol concentration. Increasing the alcohol concentration does not significantly change the absorption peak in the amide II region for random coiled material (SF-R). However, the absorption peaks of amide I and amide III are shifted, especially the absorption peak in the amide I region, which shifts to a lower wavenumber. This result shows that with the increase of alcohol content, the rearrangement of hydrogen bonds in the random coil material causes silk fibroin molecules to gradually convert from random coil to β-sheet structure [21,22]. Generally, it is difficult to directly distinguish random coils and α-helices by FTIR [13]. Combining FTIR spectra with XRD data allows us to determine that the shift of the absorption peak in the amide I region should be caused by the coexistence and superposition of α-helix, random coil and β-sheet peaks. This indicates that methanol and ethanol immersions can promote the transformation of random coil structures to α-helix and β-sheet structures (SF-R). However, the crystal structure of Silk I in SF-I samples has good stability in alcohol treatment, which is consistent with the results of XRD test.

#### 2.2.3. Raman Scattering Spectrum

To further illustrate the changes in the structure of silk fibroin materials, we examined the changes in the Raman spectra of silk fibroin materials. In the Raman spectrum (Figure 4), the silk fibroin random coil material (SF-R) has scattering peaks near 1665 cm^−1^ (amide I), 1263 cm^−1^, 1234 cm^−1^ (amide III), and 1104 cm^−1^, which is a typical random coil conformation [23,24]. It can be seen from Figure 5a that, after 8 h of 75% methanol (*v*/*v*) treatment, the amide I band of SF-R material shifted to 1669 cm^−1^ and the width became narrower. The peak at 1264 cm^−1^ in the amide III region weakened, and the peak at 1231 cm^−1^ became sharp. A sharp scattering peak was produced at 1085 cm^−1^ and the scattering peak at 1104 cm^−1^ disappeared. This indicates that 75% methanol (*v*/*v*) treatment will result in the production of β-sheet structure. In contrast, the spectra of SF-I samples (blue and pink curves) in Figure 4a remain mostly unchanged at 1661 cm^−1^ (amide I), 1273 cm^−1^, 1243 cm^−1^ (amide III), and 1106 cm^−1^, proving that Silk I crystalline structure is stable after being treated with 75% methanol (*v*/*v*). This is consistent with the previous conclusions of XRD and FTIR. Due to the influence of tyrosine residues, the silk fibroin material in Figure 4 shows a strong band at 854 cm^−1^. Both silk fibroin materials showed a band centered at 934 cm^−1^ in the region of 900~1000 cm^−1^, and the scattering peak of the Silk I material (SF-I) remained unchanged after being treated with 75% methanol (*v*/*v*) for 8 h. However, after the treatment mentioned above, the peak value of the random coiled material (SF-R) changed. In the 800~900 cm^−1^ area, the left shoulder at 854 cm^−1^ disappeared, resulting in a great reduction in the strip strength at 854 cm^−1^. Figure 4b–d shows the same pattern as Figure 4a. After treating random coiled materials (SF-R) with different concentrations of alcohol for 8 h, the spectral band of amide I shifted to a higher wavenumber, the peak shape became sharper, and the peak in the amide III region moved to 1233 cm^−1^. These phenomena indicate that the random coil conformation is transformed into a β-sheet structure [25]. However, the Raman scattering peaks of Silk I material (SF-I) remained mostly stable, without significant changes, which is consistent with the XRD and FTIR results. These results prove that Silk I crystalline in SF-I material can exist stably after methanol and ethanol treatment.

#### 2.2.4. Scanning Electron Microscope (SEM) Images

Microscopic morphology analysis by SEM (Figure 5) found that the pore size and pore arrangement density of Silk I material (SF-I) did not significantly change before and after the 90% methanol (*v*/*v*) treatment. The surface of the SF-I material has regular crank-shaped protrusions with a length of about 100~500 nm. Random nanofilaments wound around each other and connected the crank body together, forming a stable network structure [14]. As a control, the degummed natural silk fibroin fiber dominated by Silk II crystals shows a regular sheet structure, which is different from the network structure of Silk I. Meanwhile, after the 90% methanol (*v*/*v*) treatment, the surface of the random coil material (SF-R) shrinks significantly and becomes more rigid, changing the surface from flat to rough. This demonstrated that the material changes from random coil to β-sheet structure during the 90% methanol (*v*/*v*) treatment, forming Silk II crystals similar to silk fibroin fibers. These results are consistent with the XRD, FTIR, and Raman analyses. In the process of the experiment, we found that both methanol and ethanol would lead to the transformation of random crimp material to the crystal structure of Silk II, and the crystal structure of Silk I would not be affected. Since methanol treatment has a more significant effect than ethanol treatment on the random crimp structure, we provided the SEM image of the methanol-treated scaffold in the manuscript.

### 2.3. Time Stability

To explore the long-term storage stability of Silk I crystalline material, SF-I and SF-R samples were placed at room temperature for one year and tested using XRD. Figure 6 shows that there is almost no change in the XRD patterns of Silk I material (SF-I) before and after being placed for one year. X-ray diffraction peaks mainly appear at 12.2°(ms) and 19.7°(s), showing the characteristic diffraction peaks of Silk I crystal structure. This revealed that Silk I material can maintain a stable structure at room temperature for more than one year. In contrast, the random coil material (SF-R) shows a new Silk I crystal diffraction peak at 12.2° after being placed for one year. The FTIR spectra showed the same phenomenon as the XRD analysis. The absorption peaks of SF-I porous materials did not change significantly after one year, and the infrared characteristic absorption peaks were found at 1650 cm^−1^, 1532 cm^−1^, and 1237 cm^−1^. However, the SF-R materials changed from random coil structure to α-helix structure after one year.

### 2.4. Thermal Stability

To evaluate the thermal stability of the silk materials, thermogravimetric analysis was used to determine the weight loss rate of random coiled materials (SF-R), Silk I materials (SF-I), and natural silk fibroin fibers (mainly Silk II crystals) (Figure 7). Figure 7 shows the change in mass percentage of each sample during heating from room temperature to 450 °C. The mass loss percentage curve in Figure 7 reveals that each sample experienced three main stages of weight loss: In the first stage, there was a slight quality drop at 100 °C, the weight loss of random coiled material (SF-R) was 10.80%, Silk I material (SF-I) lost 9.08%, and silk fibroin fiber lost 7.02%. This part of the weight loss was caused by the volatilization of moisture in the material. The second stage is between 200 °C and 300 °C, which is a slow thermal decomposition stage, and the initial decomposition temperature is given. The initial decomposition temperatures of the three samples corresponded to 266.70 °C, 260.61 °C, and 286.37 °C, respectively. This shows that any type of silk fibroin material has good thermal stability, and there is no clear thermal degradation before 200 °C. The weight loss in the third stage was between 300 °C and 400 °C, and all samples had rapid mass loss, showing rapid weight loss. The temperature corresponding to the maximum degradation rate is 285.43 °C (SF-R), 281.38 °C (SF-I), and 313.73 °C (Silk fiber). The weight loss is mainly caused by the breakage of peptide bonds and side groups. Silk I crystal is a metastable structure, making its thermal decomposition slightly easier than that of Silk II crystal. When the temperature reaches 450 °C, the weight remaining rate of random coiled material is similar to that of the Silk I material.

### 2.5. Enzyme Degradation Stability

Figure 8 shows that in the PBS (phosphate buffered saline) solution, the quality of Silk I crystalline structure material has an exceedingly small drop of only 2.1% in 3 days, indicating that Silk I material (SF-I) has good thermal stability. When Silk I material (SF-I) was immersed in 240 U/mL α-chymotrypsin solution, the mass reduction rate within one hour reached 21.6%, and, by the time 8 h had passed, the material degradation rate reached about 42.36%. One day later, the degradation rate of Silk I material (SF-I) reached 60.75%, 68.91% on the second day, and reached 84.53% on the third day. These results yield that, under the catalysis of α-chymotrypsin, the degradation rate of Silk I material (SF-I) is much higher than its degradation rate in PBS buffer. Whether in PBS buffer or in α-chymotrypsin solution, the mass loss of silk fibroin fiber is small, and the degradation rate is less than 2%. This proves that Silk I material has the performance of rapid enzymatic degradation, while silk fiber with Silk II crystalline structure has significant resistance to enzymatic degradation.

Silk I crystalline structure material provides the possibility for the development of water-insoluble but fast enzymatically degradable biological materials. As a biomedical material, Silk II crystal structure material has poor hygroscopicity and flexibility, and is not conducive to biological degradation and absorption. Silk I crystal structure material can effectively improve the shortcomings of the Silk II crystal structure. The preparation of porous silk fibroin scaffolds with rapid biodegradable Silk I crystal structure is expected to be used as a new material for tissue engineering scaffolds, tissue induction materials, cell culture materials, and controllable drug carriers.

## 3. Discussion

Figure 9 depicts a scheme proposed to explain the structural change mechanism of various silk fibroin materials before and after treatments with methanol and ethanol. The glass transition temperature (T_g_) of the frozen silk fibroin solution is about −34~−20 °C, and the initial melting temperature (T_m_) is −8.5 °C [20]. When the silk fibroin solution was frozen below T_g_, the molecular chain of silk fibroin protein could not move freely and retained its the amorphous structure. When the freezing temperature is between T_g_ and T_m_, within a certain crystallization temperature range, the silk fibroin molecular chains will move together to form a crystal structure. In the frozen silk protein aqueous solution, the water in the system is an effective plasticizer for protein. The presence of water can significantly reduce the glass transition temperature of silk fibroin [10,26]. The freezing temperature used in this study is −7 °C with a freezing time of 15 days. Extending the freezing time results in the concentration of silk fibroin to gradually increase, promoting the folding of silk fibroin molecular chain [16,27]. This ultimately leads to a transformation of the silk fibroin conformation, and the formation of crystal structure. However, due to the high viscosity of the solution, the movement of the silk fibroin molecular chain was hindered; therefore, only Silk I crystal structure can be formed [16].

To date, several methods have been used to induce the crystallization of Silk I [7,10]. The water-insoluble silk film with Silk I structure was prepared by slow drying or adding glycerin [28]. Silk I crystalline material prepared by freeze-drying is green and environmentally friendly, without any reagents, and can exist stably for more than one year. The Silk I crystal structure material prepared in this experiment can still maintain a stable Silk I crystal structure after being soaked in methanol and ethanol for 1, 2, 4, and 8 h. This conclusion provides a solid theoretical basis for Silk I medical treatment. Degradation behavior is crucial in the field of biomaterials. Silk I crystalline material as a biomedical material should study this characteristic. Previous studies show the degradation of silk fibroin films based on Silk I was studied using protease XIV solution [10,14,29]. The water insolubility and enzyme degradable properties of the Silk I crystalline structure is confirmed by the results found. Silk I crystal has significant thermal stability and thermal decomposition that occurs at 260.61 °C, while the silk fibroin fibers mainly composed of Silk II decompose at 286.37 °C. After immersion treatment, the materials with random coil structure can be transformed into β-sheet structure, making the material become rough and rigid, which is consistent with the previous conclusion [30,31].

At present, water-insoluble silk fibroin membrane and porous scaffold with Silk I structure have been developed [7,10,20,28]. However, as a potential medical product, the shelf-life is a very important indicator. It determines whether the product is convenient to store and sell. In addition, the sterilization method of medical product or devices is also an important content. General sterilization methods include high temperature sterilization, ethanol sterilization and ethylene oxide sterilization. It is generally not recommended to use ethylene oxide sterilization to prevent the residue of ethylene oxide for the porous material. In hospitals, ethanol is commonly used as convenient sterilization method. Therefore, it is necessary to investigate the ethanol, thermal and time stability of Silk I-based scaffold if it is used as a medical device. The results in this article found that Silk I-based scaffold is stable in alcohols, such as methanol and ethanol. In addition, Silk I-based scaffold can be sterilized with ethanol, and it is also stable under high temperature or long-term storage. These advantages indicate that Silk I-based scaffold materials will have a wide range of applications in the near future.

## 4. Materials and Methods

### 4.1. Experimental Materials

Fresh mulberry silkworm cocoons of Bombyx mori from Xiancan Silk Biotechnology Co. Ltd. (Suzhou, China), dialysis bags from Puyi Biotechnology Co. Ltd. (Shanghai, China), sodium carbonate, sodium bicarbonate, and lithium bromide from Tiancheng Chemical Co. Ltd. (Shandong, China), anhydrous methanol and absolute ethanol from Aladdin Reagent Co. Ltd. (Shanghai, China), and α-chymotrypsin from McKellin Biochemical Technology Co. Ltd. (Shanghai, China) were purchased for this study.

### 4.2. Preparation of Silk Fibroin Solution

Eighty grams of silkworm cocoon shells were weighed and steamed in a 0.3% (*w*/*v*) Na_2_CO_3_/0.1% (*w*/*v*) NaHCO_3_ solution at 100 °C three times, each for half an hour, stirring once every ten minutes. To remove the degraded sericin on the surface, deionized water was used to rinse the bombyx mori silk after each cooking. The silk was then dried of the water absorbed on the surface of the silk fibroin at 60 °C. The weighed 15 g of degummed silk fibroin was then dissolved in a LiBr solution (9.3 M) at 65 °C for 1 h. After cooling, the dissolving system was placed in deionized water at 4 °C for dialysis for 3 to 4 days. Particulate impurities in the solution were then filtered to remove the resulting silk fibroin solution.

### 4.3. Preparation of Silk Fibroin Materials

#### 4.3.1. Preparation of Silk I Crystalline Structure Materials

The silk fibroin (SF) solution (6.0%, *w*/*v*) was poured into a polyethylene petri dish, and then treated at a temperature of −7 °C for more than 15 days. The frozen samples were freeze-dried in a Christ freeze dryer for 48 h to obtain a porous material with Silk I crystalline structure (SF-I in Figure 10A). The forming mechanism of SF-I is explained in detail in Section 3.

#### 4.3.2. Preparation of Random coil Structure Materials

The silk fibroin (SF) solution (6.0%, *w*/*v*) was poured into a polyethylene petri dish, frozen at temperature of −40 °C for 2 h, and freeze-dried it with a freeze dryer for 48 h to obtain a porous material with a curled structure (SF-R in Figure 10B).

### 4.4. Water Resistance Stability Determination

After drying the silk fibroin material in a constant temperature and humidity environment (25 °C, relative humidity 65%) for 24 h, part of the samples were cut, weighed, denoted as M_1_, and then dried in an oven at 105 °C to a constant weight, denoted as M_2_. The moisture content of the silk fibroin porous material was calculated according to Equation (1). One-tenth of a gram of each sample was placed in a 10 mL centrifuge tube, labeled as M_0_. Each sample was shaken with deionized water at a bath ratio of 1:100 in a 37 °C water bath for 12 h. After shaking, the samples were centrifuged at a speed of 3500 r/min for 15 min, and the supernatant was taken to measure the value of absorbance A on a Smartspec ultraviolet spectrophotometer (λ = 278 nm). The hot water loss rate of silk fibroin material was calculated in regards to Equation (2). Three parallel samples were in each group, and the results were averaged.
(1)$$Moisture\;content(\%)=\frac{M_{1}-M_{2}}{M_{1}}\times 100\%,$$
(2)$$Dissolution\;\mathrm{rate}(\%)=\frac{\mathrm{KAV}}{M_{0}{-M}_{0}\times \mathrm{Moisture}\;\mathrm{content}}\times 100\%,$$
where K is the UV absorption constant of the silk fibroin solution (K = 1.1012), A is the absorbance, and V is the total volume of the solution.

### 4.5. Determination of Stability of Monohydric Alcohol Treatment

#### 4.5.1. Methanol Treatment

To explore the stability of the crystal structure of silk fibroin material Silk I, two samples (SF-I and SF-R) were immersed in 99%, 90%, and 75% (*v*/*v*) methanol, respectively The processing time gradient was set to 1 h, 2 h, 4 h, 8 h respectively. After soaking, the samples were placed in a constant temperature and humidity room (25 °C, relative humidity 65%) to dry for 24 h. The experimental treatment conditions are based on previous studies by Puerta et al. [31].

#### 4.5.2. Ethanol Treatment

The SF-I and SF-R samples were immersed in 99%, 90%, and 75% (*v*/*v*) ethanol solution, respectively, with a time gradient set for 1 h, 2 h, 4 h, and 8 h, respectively. Following immersion treatment each sample was dried in a constant temperature and humidity room (25 °C, relative humidity 65%) for 24 h. The experimental treatment conditions are based on previous studies Puerta et al. [31].

### 4.6. Long Time Stability

To understand the long-term storage stability of Silk I crystalline material, both SF-I and SF-R samples were placed at room temperature for 1 year, after which the structural changes were measured.

### 4.7. Thermal Stability Determination

TA Differential Thermometer SDT Q600 (TA Instruments, New Castle, DE, USA) was used to measure the mass stability of the silk samples with increasing temperature. The thermogravimetric curve was obtained at a gas flow rate of 50 mL/min under a nitrogen atmosphere. Four milligrams of each sample was heated from 25 °C to 450 °C at a heating rate of 10 °C/min. During heating, the percentage of mass change was recorded.

### 4.8. Enzymatic Degradation

Each sample was cut into 5 mm × 5 mm size, and the initial weight was recorded as M_0_. The silk fibroin sample was then put into a 240 U/mL α-chymotrypsin solution (pH 7.3) for degradation [32]. The temperature was set at 37 °C, with a time gradient of 1 h, 2 h, 4 h, 8 h, 1 d, 2 d, 3 d. After degradation, the sample was taken out, rinsed thoroughly with deionized water, and dried in an oven at 80 °C until the weight remained constant, mass M. The remaining mass percentage of the sample was calculated by the formula (M/M_0_ × 100%). The results of the three parallel samples were averaged. The samples treated with enzyme-free phosphate buffer were used as controls. All silk fibroin materials were processed using the same procedure.

### 4.9. Structural Characterization

Processed and unprocessed samples were cut into pieces with scissors, and an 80-mesh sieve was used to collect the finely divided samples.

#### 4.9.1. X-ray Diffraction Analysis

The fully automatic X’PERT PRO MPD ray diffractometer (Bruker Corporation, Berlin, Germany) was used to test the structure of each sample. The diffraction intensity curve was recorded between 5° and 45° under the conditions of 10°/min scanning speed, 40 kV, and 30 mA.

#### 4.9.2. Fourier Transform Infrared Spectroscopy

Each sample was thoroughly mixed with KBr, crushed, and then compressed into transparent small discs with a tablet press. A Nicolet 5700 infrared spectrum analyzer (Thermo Nicolet Corporation, Waltham, MA, USA) was used to analyze the infrared absorption spectrum of the sample with a scanning step range of 400~4000 cm^−1^.

#### 4.9.3. Raman Scattering Spectroscopy

Raman spectrum was measured using a Japanese HORIBA Raman Microscopy (HORIBA Ltd., Kyoto City, Japan). The excitation wavelength was 532 nm, slit width was 100 μm, and 1200 gr/mm grating was selected. The scanning time of the fixed sample was 20 s, and the Raman scattering spectrum recording step range was 200~2000 cm^−1^.

### 4.10. Morphology Characterization

The lyophilized silk fibroin samples before and after the 90% methanol immersion treatment were pasted with carbon conductive glue on a dedicated sample table, and Au was sprayed for 180 s under 10 mA conditions to give the samples surface conductivity. The morphology of samples were characterized and analyzed by S4700 cold field emission scanning electron microscope.

## 5. Conclusions

Comparing different stability performance of silk fibroin material with random coil structure, silk fiber with Silk II structure, and Silk I-based scaffold, it is found that Silk I-based scaffold has significant water resistance, thermal stability, and time stability. The immersion treatment of molecular alcohols also maintained adequate chemical stability. The soaking of methanol and ethanol can promote the conversion of silk fibroin from random coil structure to Silk II structure, but it will not cause the change of Silk I crystal. Silk I crystal structure can exist stably in methanol and ethanol solution. Silk I-based scaffold can be quickly degraded by enzyme. This kind of material with stable structure and fast enzymatic hydrolysis provides a new choice for the application of silk fibroin in the biomedical field.

## Abbreviation

SF Silk FibroinSF-R Silk Fibroin Random coil porous materialSF-ISilk Fibroin Silk I crystalline structure porous materialGAGAGSAmino acid sequence Gly-Ala-Gly-Ala-Gly-SerXRDX-ray diffractionFTIRFourier transform infrared spectroscopySEMScanning electron microscopeTGAThermogravimetric analysisMTMethanol treatment.TgGlass transition temperatureTmMelting temperature 

## Figures and Tables

**Figure 1 ijms-22-04136-f001:**
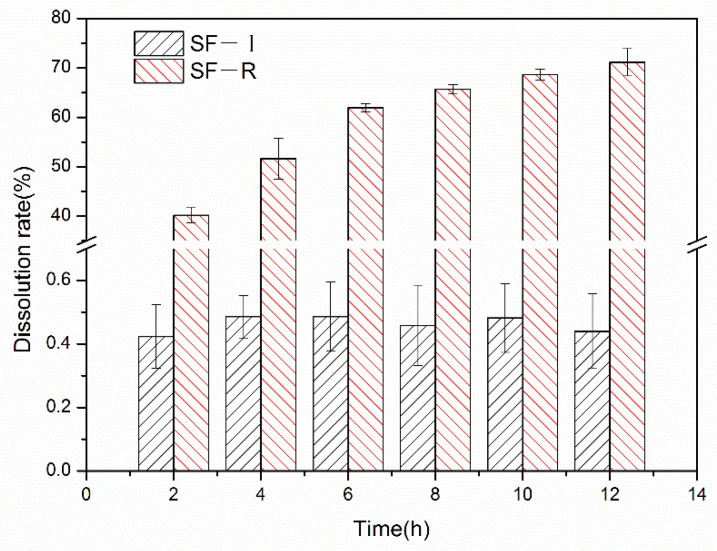
Water loss rate of different silk fibroin materials (SF-I and SF-R) for 2, 4, 6, 8, 10, and 12 h.

**Figure 2 ijms-22-04136-f002:**
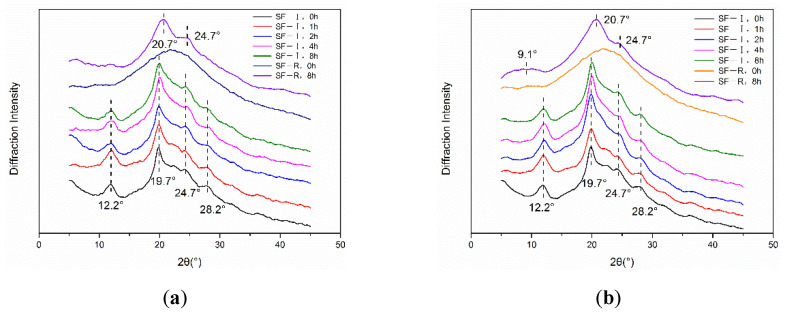
X-ray diffraction (XRD) analysis of SF-I and SF-R materials before and after alcohol treatment: (**a**) 75% methanol, (**b**) 90% methanol, (**c**) 99% methanol, (**d**) 75% ethanol, (**e**) 90% ethanol, (**f**) 99% ethanol.

**Figure 3 ijms-22-04136-f003:**
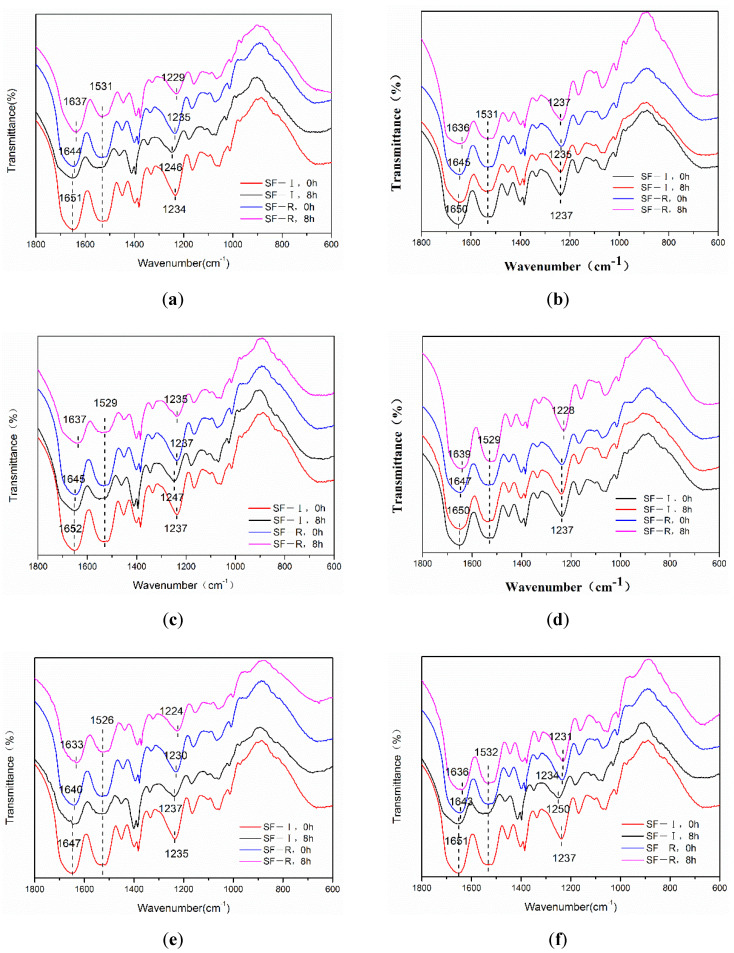
Fourier transform infrared spectroscopy (FTIR) spectra of SF-I and SF-R materials before and after alcohol treatment: (**a**) 75% methanol, (**b**) 90% methanol, (**c**) 99% methanol, (**d**) 75% ethanol, (**e**) 90% ethanol, (**f**) 99% ethanol.

**Figure 4 ijms-22-04136-f004:**
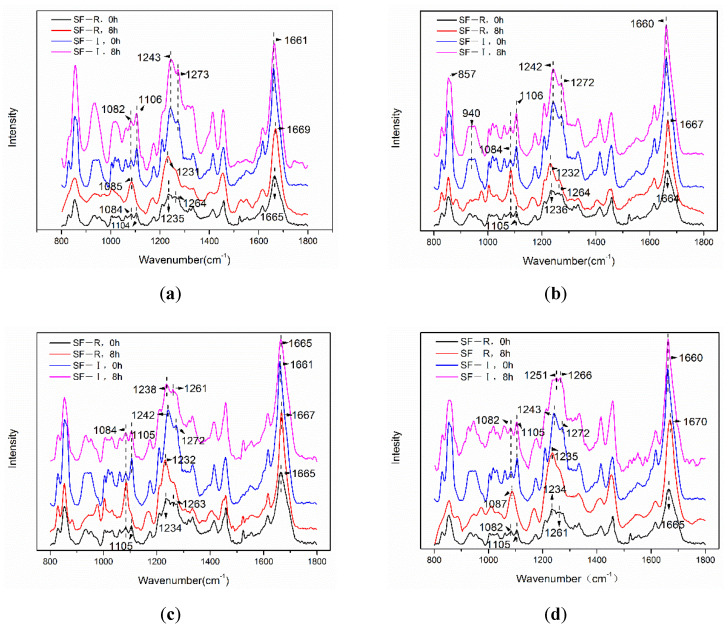
Raman spectra of silk fibroin materials (SF-I and SF-R) before and after alcohol treatments: (**a**) 75% methanol, (**b**) 90% methanol, (**c**) 75% ethanol, (**d**) 90% ethanol.

**Figure 5 ijms-22-04136-f005:**
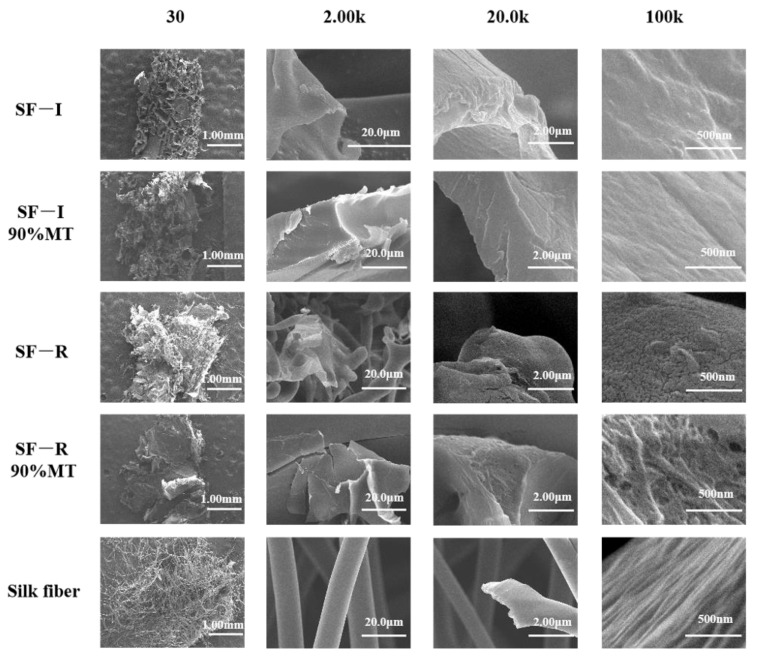
Scanning electron microscope (SEM) images of silk fibroin materials (SF-I and SF-R) before and after 90% methanol (*v*/*v*) treatment.

**Figure 6 ijms-22-04136-f006:**
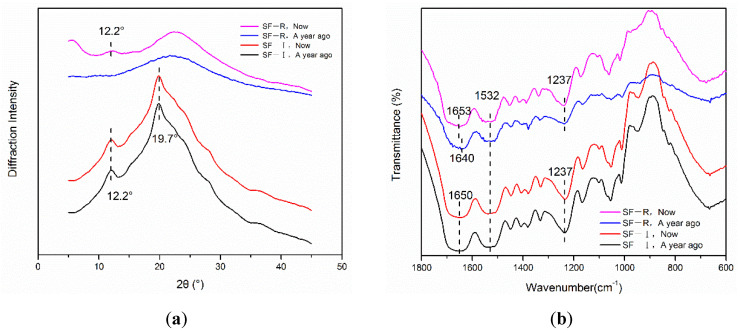
Silk fibroin materials (SF-I and SF-R) before and after being placed at room temperature for one year: (**a**) XRD analysis, (**b**) FTIR spectra.

**Figure 7 ijms-22-04136-f007:**
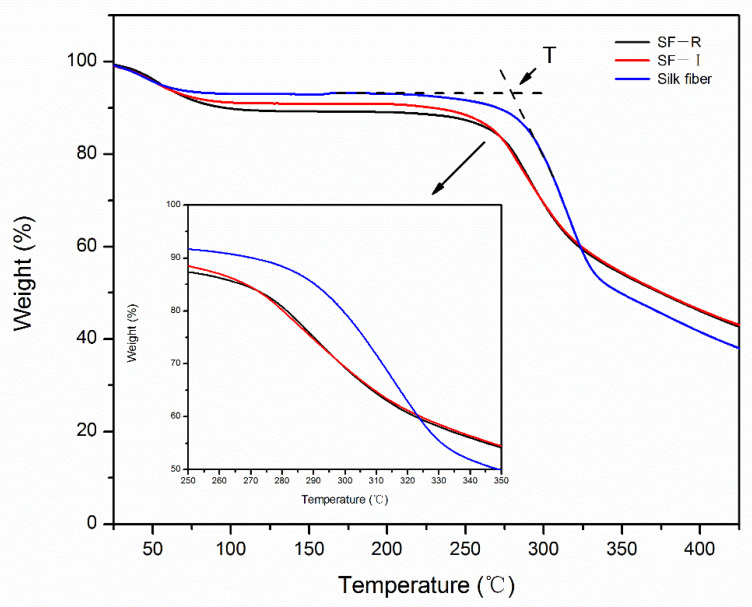
Thermogravimetric analysis (TGA) curves of different silk fibroin materials (SF-I, SF-R, and silk fibers).

**Figure 8 ijms-22-04136-f008:**
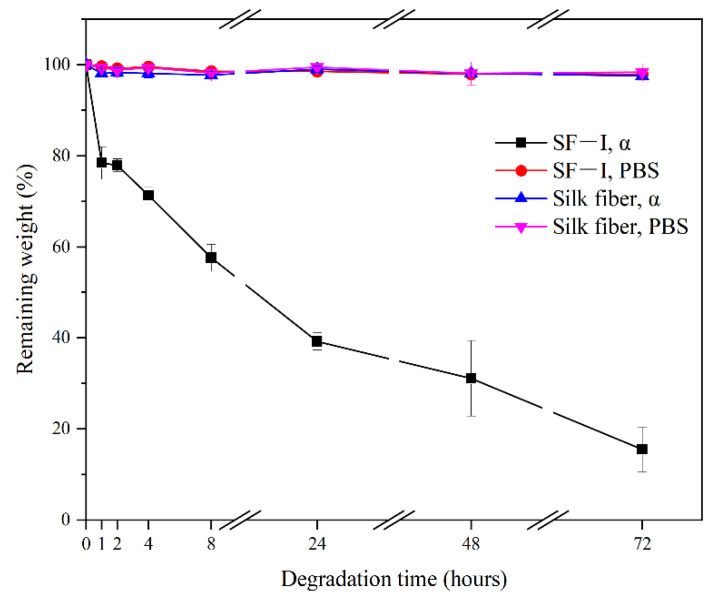
Enzymatic and PBS degradation curves of different silk fibroin materials (SF-I and silk fibers).

**Figure 9 ijms-22-04136-f009:**
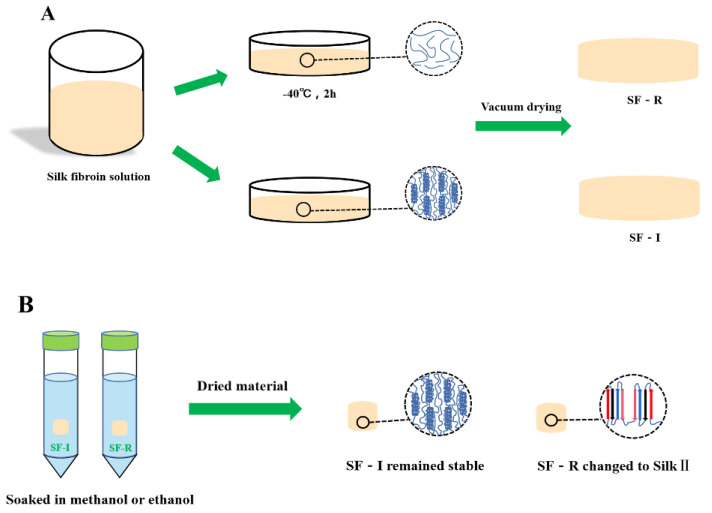
(**A**) Formation mechanism of SF-R and SF-I materials, and (**B**) structure of silk fibroin materials (SF-R and SF-I) after methanol or ethanol treatments.

**Figure 10 ijms-22-04136-f010:**
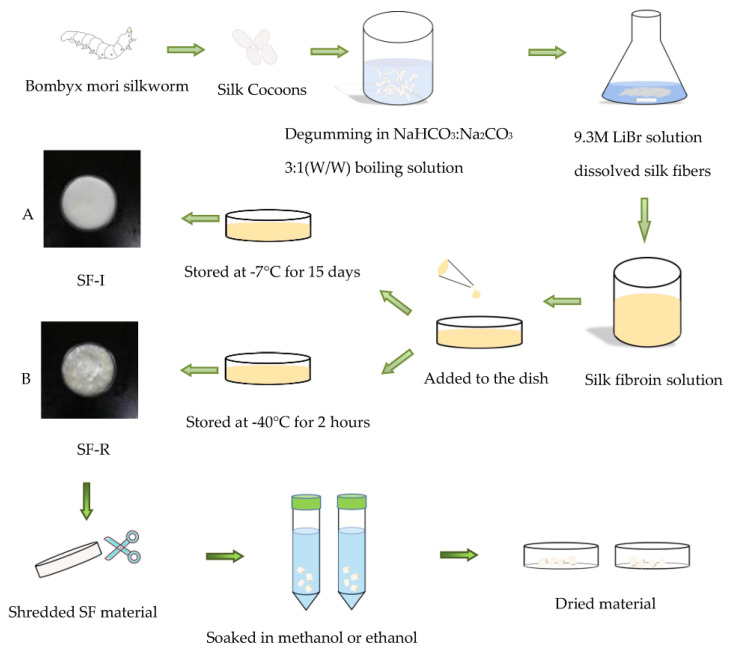
Experimental procedure to prepare (**A**) Silk I crystalline structure dominated silk materials, and (**B**) uncrystallized (randomly coils dominated) silk materials.

## Data Availability

Not applicable.

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
