# Peer review of "Chemical, Thermal, Time, and Enzymatic Stability of Silk Materials with Silk I Structure"

_ijms, 2021, doi:10.3390/ijms22084136_

Round 1
Reviewer 1 Report
In this article, authors evaluated the stability of silk fibroin materials. The article resulted as very interesting because open the possibility to obtain stable fibroin-based scaffold without the necessary conformational change from Silk I to Silk II.
I recommended some revisions.
Comments:
- I suggest to add a graphical abstract to better explain all study steps. I also suggest to add a list of all abbreviations used in the manuscript.
- Which are the main advantages allowed by the use of Silk I fibroin with respect to Silk II fibroin?
- SEM images showed only the scaffold after methanol treatment, which are the results obtained with ethanol treatment?
- It is not clear why the different freeze temperature (Figure 10) influences the fibroin scaffold. Please explain.
- In the conclusion section could be interesting to underline why the maintenance of Silk I structure is too important for the fibroin scaffold properties. I also suggest to add in conclusion a summary of the experimental conditions that allowed to obtain a stable Silk I-based scaffold.
Author Response
We are thankful to reviewer for critical reading, thoughtful comments and suggestions on our previously submitted manuscript. We have carefully taken into consideration all the comments for preparing our revised manuscript. For general comments to the authors, we are also grateful to the Reviewer for further advice and suggestions to improve the manuscript. We have responded to the reviewer’s comments as stated below:
Reviewer: 1
In this article, authors evaluated the stability of silk fibroin materials. The article resulted as very interesting because open the possibility to obtain stable fibroin-based scaffold without the necessary conformational change from Silk I to Silk II.
I recommended some revisions.
- I suggest to add a graphical abstract to better explain all study steps. I also suggest to add a list of all abbreviations used in the manuscript.
Thank you for your comments. We have added a graphical abstract (based on Figure 9) to better explain all the experimental steps. The specific significance of related abbreviations is also added in the present manuscript.
List of abbreviation
|
SF |
Silk Fibroin |
|
SF-R |
Silk Fibroin Random coil porous material |
|
SF-I |
Silk Fibroin Silk I crystalline structure porous material |
|
GAGAGS |
Amino acid sequence Gly-Ala-Gly-Ala-Gly-Ser |
|
XRD |
X-ray diffraction |
|
FTIR |
Fourier transform infrared spectroscopy |
|
SEM |
Scanning electron microscope |
|
TGA |
Thermo gravimetric analysis |
|
MT |
Methanol treatment. |
|
Tg |
Glass transition temperature |
|
Tm |
Melting temperature |
- Which are the main advantages allowed by the use of Silk I fibroin with respect to Silk II fibroin?
Thank you for your questions. Compared with Silk II crystal, Silk I crystal material has better hydrophilic property, flexibility, and has quick enzymic degradability. These advantages are conducive to biological degradation and absorption. Its good flexibility can make up for the shortcoming that Silk II crystalline structure material is hydrophobic and fragile.
- SEM images showed only the scaffold after methanol treatment, which are the results obtained with ethanol treatment?
Thank you for your questions. In the process of the experiment, we found that both methanol and ethanol would lead to the transformation of the random coil structure to a crystal structure of Silk II, and that the crystal structure of Silk I would not be affected. Since the methanol treatment has a more significant effect than the ethanol treatment on the random coil structure, we provided the SEM image of the methanol treated scaffold in the manuscript.
- It is not clear why the different freeze temperature (Figure 10) influences the fibroin scaffold. Please explain.
Thank you for your questions. As we discussed in the Discussion section: “The glass transition temperature (Tg) of the frozen silk fibroin solution is about -34 ~ -20 oC, and the initial melting temperature (Tm) is -8.5 oC[17]. When the silk fibroin solution was frozen below Tg, the molecular chain of silk fibroin protein could not move freely and retained its the amorphous structure. When the freezing temperature is between Tg and Tm, within a certain crystallization temperature range, the silk fibroin molecular chains will move together to form a crystal structure. In the frozen silk protein aqueous solution, the water in the system is an effective plasticizer for protein. The presence of water can significantly reduce the glass transition temperature of silk fibroin[10, 23]. The freezing temperature used in this study is -7 oC with a freezing time of 15 days. Extending the freezing time results in the concentration of silk fibroin to gradually increase, promoting the folding of silk fibroin molecular chain[24, 25]. This ultimately leads to a transformation of the silk fibroin conformation, and the formation of crystal structure. However, due to the high viscosity of the solution the movement of the silk fibroin molecular chain was hindered, therefore only Silk I crystal structure can be formed [25].”
- In the conclusion section could be interesting to underline why the maintenance of Silk I structure is too important for the fibroin scaffold properties. I also suggest to add in conclusion a summary of the experimental conditions that allowed to obtain a stable Silk I-based scaffold.
Thank you for your comments. We added a paragraph for why the maintenance of Silk I structure is too important for the fibroin scaffold properties in the section of Discussion. “At present, water-insoluble silk fibroin membrane and porous scaffold with Silk I structure have been developed[7, 10, 17, 26]. However, as a potential medical product, the shelf-life is a very important indicator. It determines whether the product is convenient to store and sell. In addition, the sterilization method of medical product or devices is also an important content. General sterilization methods include high temperature sterilization, ethanol sterilization and ethylene oxide sterilization. It is generally not recommended to use ethylene oxide sterilization to prevent the residue of ethylene oxide for the porous material. In hospitals, ethanol is commonly used as convenient sterilization method. Therefore, it is necessary to investigate the ethanol, thermal and time stability of silk I-based scaffold if it is used as a medical device. The results in this article found that Silk I-based scaffold is stable in alcohols such as methanol and ethanol. In addition, Silk I-based scaffold can be sterilized with ethanol and it is also stable under high temperature or long-term storage. These advantages indicate that Silk I-based scaffold materials will have a wide range of applications in the near future.”
In the conclusion section, we have also given a summary of stability of silk I-based scaffold: “Comparing the different stability performances of silk fibroin material with the random coil structure, silk fiber with Silk II structure and Silk I-based scaffold, it is found that Silk I-based scaffold has significant water resistance, thermal stability and time stability. The immersion treatment of molecular alcohols also maintained adequate chemical stability”.
Reviewer 2 Report
Authors have done an extensive biophysical analysis of Silk materials with Silk I Structure.
Comments:
1) Title: Instead of Time, words like shelf-life could be used.
2) Introduction:
- a) In the last few lines of the introduction, the application of improved silk materials in biomedical field is mentioned. Can author provide some examples with references?
3) Results
- a) 2.1: What does author mean when they say that the sample is lost during dissolution process? Has there been any secondary structure determination from Circular Dichroism done?
Author Response
Reviewer: 2
Authors have done an extensive biophysical analysis of Silk materials with Silk I Structure.
Comments:
- Title: Instead of Time, words like shelf-life could be used.
Thank you for your comments. It is really a Time Stability test. Shelf-life means the length of time for which an item remains usable, fit for consumption, or saleable. However, in this paper, we only test the Time Stability of materials, but not the end products for consumer use. For this reason, we choose to select the word ‘time’ and not the phrase ‘shelf-life’. To better demonstrate the importance of ‘shelf-life’ for the potential product in the future, we also added a paragraph for the fibroin scaffold properties in the section of Discussion:
“At present, water-insoluble silk fibroin membrane and porous scaffold with Silk I structure have been developed[7, 10, 17, 26]. However, as a potential medical product, the shelf-life is a very important indicator. It determines whether the product is convenient to store and sell. In addition, the sterilization method of medical product or devices is also an important content. General sterilization methods include high temperature sterilization, ethanol sterilization and ethylene oxide sterilization. It is generally not recommended to use ethylene oxide sterilization to prevent the residue of ethylene oxide for the porous material. In hospitals, ethanol is commonly used as convenient sterilization method. Therefore, it is necessary to investigate the ethanol, thermal and time stability of silk I-based scaffold if it is used as a medical device. The results in this article found that Silk I-based scaffold is stable in alcohols such as methanol and ethanol. In addition, Silk I-based scaffold can be sterilized with ethanol and it is also stable under high temperature or long-term storage. These advantages indicate that Silk I-based scaffold materials will have a wide range of applications in the near future.”
- Introduction: a) In the last few lines of the introduction, the application of improved silk materials in biomedical field is mentioned. Can author provide some examples with references?
Thank you for your comments. We have added the application of silk fibroin modified material in the biomedical field in the present manuscript.
“Lu S[16] inserted the prepared porous material Silk I into the subcutaneous layer on the dorsal surface of SD rats, and all the rats could survive and live in good condition. After 6 weeks, the porous material Silk I was degraded and absorbed by the rats at most. Hu X et al. demonstrated a new physical approach to control the structure of fibrous proteins through temperature-controlled water vapor annealing (TCWVA), making it a suitable starting material for any complex format such as patterned films, foams, nanofibers, micro/nano-particles, or gels[10]. Jin H effectively controlled the β-sheet ratio to obtain a water-stable film containing Silk I crystals[17]. Zhu M et al. used combined silk I microneedles for drug delivery of insulin delivery[18].”
- Results a) 2.1: What does author mean when they say that the sample is lost during dissolution process? Has there been any secondary structure determination from Circular Dichroism done?
Thank you for your questions. Results 2.1 is to determine the dissolution rate of silk fibroin porous materials in hot water. The random coil structure of silk fibroin is dissolved in water and becomes silk fibroin solution. It was found that the sample lost some weight during dissolution process. This is due to a small amount of random coil structure in the SF-I material, which is dissolved in water. However, the random coil structure porous material SF-R contains a large number of amorphous structures, so its dissolution rate can reach 71% in 12 hours.
We test the secondary structure by FTIR and Raman spectra, but did not use Circular Dichroism in this paper. As for Circular Dichroism, the materials should have light transmittance, it is used to test secondary structure of solution. It is difficult to test the secondary structure of solids, especially for porous materials in this study.
Round 2
Reviewer 1 Report
Authors modified the manuscript as requested.